# Understanding Salinity-Driven Modulation of Microbial Interactions: Rhizosphere versus Edaphic Microbiome Dynamics

**DOI:** 10.3390/microorganisms12040683

**Published:** 2024-03-28

**Authors:** Rui Li, Haihua Jiao, Bo Sun, Manjiao Song, Gaojun Yan, Zhihui Bai, Jiancheng Wang, Xuliang Zhuang, Qing Hu

**Affiliations:** 1Research Center for Eco-Environmental Sciences, Chinese Academy of Sciences, Beijing 100085, China; ruili_st@rcees.ac.cn (R.L.); jiaohaihua68@163.com (H.J.); sunbo3151@163.com (B.S.); songmanjiao19@mails.ucas.ac.cn (M.S.); gjyan_st@rcees.ac.cn (G.Y.); zhbai@rcees.ac.cn (Z.B.); xlzhuang@rcees.ac.cn (X.Z.); 2University of Chinese Academy of Sciences, Beijing 100049, China; 3Binzhou Institute of Technology, Weiqiao-UCAS Science and Technology Park, Binzhou 256606, China; wangjiancheng@wqucas.com; 4Department of Biological Sciences and Technology, Changzhi University, Changzhi 046011, China; 5Institute of Tibetan Plateau Research, Chinese Academy of Sciences, Beijing 100101, China; 6Xiongan Innovation Institute, Xiongan New Area, Baoding 071000, China

**Keywords:** saline soil, community assembly, rhizosphere, fungal–bacterial interactions, bipartite network, co-occurrence

## Abstract

Soil salinization poses a global threat to terrestrial ecosystems. Soil microorganisms, crucial for maintaining ecosystem services, are sensitive to changes in soil structure and properties, particularly salinity. In this study, contrasting dynamics within the rhizosphere and bulk soil were focused on exploring the effects of heightened salinity on soil microbial communities, evaluating the influences shaping their composition in saline environments. This study observed a general decrease in bacterial alpha diversity with increasing salinity, along with shifts in community structure in terms of taxa relative abundance. The size and stability of bacterial co-occurrence networks declined under salt stress, indicating functional and resilience losses. An increased proportion of heterogeneous selection in bacterial community assembly suggested salinity’s critical role in shaping bacterial communities. Stochasticity dominated fungal community assembly, suggesting their relatively lower sensitivity to soil salinity. However, bipartite network analysis revealed that fungi played a more significant role than bacteria in intensified microbial interactions in the rhizosphere under salinity stress compared to the bulk soil. Therefore, microbial cross-domain interactions might play a key role in bacterial resilience under salt stress in the rhizosphere.

## 1. Introduction

Soil salinization poses a significant challenge to global biodiversity and food security [1]. Increased salinity levels can adversely affect soil health, reducing its fertility and limiting plant growth [2]. This, in turn, impacts ecosystem dynamics and agricultural productivity [3]. For instance, studies have shown that in regions where soil salinity is high, such as certain coastal areas or arid regions, crop yields can decrease by up to 50% or more [4]. Additionally, soil salinization can lead to the loss of biodiversity, as many plant and microbial species are unable to tolerate high salt concentrations [5]. This loss of biodiversity can have cascading effects on ecosystem functioning, including nutrient cycling and soil development [6]. Moreover, salinization can exacerbate soil erosion and desertification, further compromising soil quality and agricultural viability [7].

A recent United Nations survey indicates that the global area of saline soils has exceeded 424 million hectares, with a growth rate of >3 hectares per minute on Earth [8,9]. Continuous salinization would cause serious ecological problems exemplified by reduction in microbial diversity and shifts in cross-domain interactions in different niches and ecological settings [10,11,12]. For instance, Morrissey EM et al. sampled eight tidal wetlands ranging from freshwater to oligohaline in four rivers near Chesapeake Bay to explore salinity effects on microbial community composition [13]. Additionally, a study explored both soil and sediment samples collected along a 140 m transect from the hypersaline lake La Sal del Rey to investigate the effects of environmental parameters, especially salinity, on microbial diversity [14]. Sediment samples collected in Qinghai–Tibetan lakes were also subjected to unveil the role of salinity in shaping microbial diversity and community structure [15].

High-salinity soil inhibits the relative abundance of bacteria and functional genes, affecting soil carbon metabolism, reducing soil carbon accumulation, and weakening soil carbon sink capacity [16]. Soil salinization decreases the abundance of certain bacteria, significantly hinders the diversity of soil microorganisms, and impairs soil microbial functions [17]. The soil microbiome, which plays an essential role in maintaining regional ecosystem health and stability, is sensitive to environmental stress. Changes in the soil microbiome, in turn, affect the functionality of the whole ecosystem [18,19,20,21].

Nonetheless, despite its importance as one of the most active niches within the Earth’s surface, the rhizosphere has remained relatively neglected in discussions concerning salinization. The rhizosphere is impacted by processes related to root exudation and root associations, forming a dynamic microenvironment within the soil [22,23,24]. A myriad of studies have unraveled the community assembly mechanisms and interaction relationships of microbes in the rhizosphere environment [25,26,27,28]. Microorganisms colonizing the rhizosphere exhibit intensive interactions, and their co-occurrence networks usually form a complex and stable structure, which is not easily disturbed by external factors relative to that of the surrounding bulk soils [29,30,31]. Therefore, how salinity drives microbial dynamics in the rhizosphere in terms of diversity, co-occurrence, and cross-domain interactions, as well as the synergistic effect with other environmental factors, are essential to fully understanding the microbial modulation under salt stress.

Here, bulk and rhizosphere soil samples were collected from a heavily salinized area in Xinjiang province, China, and analyzed the microecological characteristics of bacterial and fungal communities. The objectives of this study were to elucidate the dynamics of soil microbiomes in salt-stressed ecosystems and to ascertain the interactions between rhizosphere fungi and bacteria under salt stress and their distinctions from non-rhizosphere microorganisms.

## 2. Materials and Methods

### 2.1. Geological Setting and Soil Sampling

The study area is located at a typical saline botanical garden (45°26′10″ N, 85°0′20″ E) in Xinjiang province, China, established by the Xinjiang Institute of Ecology and Geography, Chinese Academy of Sciences. Situated in the piedmont alluvial slope plains of the Altay Mountains in the northern part of Karamay City, the study area comprises gravel, sand, and sandy soil of varying thicknesses (Figure 1). Local features include erosional gullies, with terrain sloping from northwest to southeast at an inclination of 2°, with elevations ranging from 379 to 485 m above sea level. The predominant surface landscape consists of a gravel desert characterized by arid and super-arid shrubs, along with salt meadows [32].

The study area falls under a temperate continental arid climate and has the following key characteristics. (1) low rainfall and large temperature fluctuations: the average annual precipitation is 105 mm, with an average annual evaporation of 3009 mm. The mean annual temperature is 8.1 °C, with scorching summers, chilly winters, and extreme temperatures ranging from −40.5 °C to 43.6 °C. (2) Weak soil, primarily consisting of gravel and sandy soils, with localized occurrences of saline meadow soils. Soil nitrogen, phosphorus, potassium, and organic matter content are generally low. Severe salinization and alkalization of the soil surface are common, along with the presence of clay salt layers and cemented layers within the soil profile. Soil salinity accumulation, compaction, and low nutrient availability significantly impede plant growth. (3) Limited water resources: surface water resources are mainly derived from temporary floods during heavy rainfall periods. Groundwater typically lies beyond 7 m of depth, posing challenges for direct absorption and utilization by desert plants. Water usage in the region heavily relies on water extraction through hydraulic facilities. The predominant vegetation in the sampling area comprises xerophytic salt-tolerant plants, which are commonly constrained by harsh ecological conditions such as drought, soil coarseness, and high salinity. The most abundant families of halophytes in the sampling area are *Amaranthaceae*, *Fabaceae*, *Asteraceae*, and *Poaceae* [33].

42 sampling points within a salt gradient (Figure 1) were randomly set up to collect non-rhizosphere and rhizosphere soil samples as described previously [34]. The soil samples were prepared for analysis by aseptically removing any visible roots and stones. Once this was performed, the samples were passed through a 2 mm sieve and then divided into two separate parts. One part of the soil was air-dried to allow for physicochemical analysis, while the other part was promptly stored at a temperature of −80 °C for future DNA extraction.

### 2.2. Soil Physicochemical Analysis

Soil electrical conductivity (EC) and pH were measured in 1:2.5 (*w*/*v*) soil–water extracts using an electrical conductivity meter (INESA, Shanghai, China) and a pH meter (INESA, Shanghai, China), respectively. Soil water content (SWC) was determined by the gravimetric method with oven drying at 105 °C for 24 h [35]. Soil organic matter (SOM) in soil samples was quantified using the loss on ignition method [13]. Total soil nitrogen (TN) was measured using the Modified Kjeldahl method [36]. Available nitrogen (AN) was determined using an AA3 continuous-flow analysis system (SEAL Autoanalyzer 3 HR, SEAL analytical Inc., Mequon, WI, USA) [37]. Total phosphorus (TP) was determined by the alkali fusion–Mo-Sb anti-spectrophotometric method [38]. Available phosphorus (AP) was determined by the sodium hydrogen carbonate solution–Mo-Sb anti-spectrophotometric method [39]. Total potassium (TK) was determined by the acid-soluble atomic emission photometric method [40]. Available potassium (AK) was determined by the flame photometer method [41].

### 2.3. Soil DNA Extraction and Bioinformatics Analysis

The FastDNA SPIN Kit for Soil was utilized to extract total genomic DNA from soil samples (MP Biochemicals, Solon, OH, USA). To amplify the hypervariable V3–V4 region of the bacterial 16S ribosomal RNA genes, primer pairs 338F/806R were employed. Similarly, to amplify the internal transcribed spacer 1 region of the fungal 18S ribosomal RNA genes, primer pairs ITS1/ITS2 were utilized [42,43].

High-throughput amplicon sequencing was performed on the Miseq PE300 platform (Illumina Inc., San Diego, CA, USA). The raw sequences were stitched, primer-removed, quality-controlled, and de-redundant using USEARCH v10 [44] and QIIME 2-2022.11 [45]. The resulting sequences were denoised and assigned to amplicon sequence variants (ASVs) using UNOISE3 in USEARCH v10. To annotate taxonomic information for bacteria and fungi, species annotators were trained using QIIME 2-2022.11 based on the SILVA 138 (https://www.arb-silva.de/documentation/release-138/, accessed on 1 October 2022) and UNITE (https://unite.ut.ee/, accessed on 1 October 2022) databases, respectively [46]. The feature table of ASVs was rarefied before calculating microbial diversity. The deposited raw sequencing data can be found in the NCBI database (https://www.ncbi.nlm.nih.gov/, accessed on 1 May 2022) with accession number PRJNA831035.

### 2.4. Statistical Analysis

To portray the fluctuating variations of microbial community attributes under the influence of salt-induced stress, the samples were categorized into eight distinct groups (N1–N4 for soils not pertaining to the rhizosphere, R1–R4 for rhizosphere soils) according to the degree of salinity and the characteristics of the soil environment (Appendix A). To determine the significance of physicochemical differences between non-rhizosphere and rhizosphere soils, we performed Wilcoxon rank-sum tests in R v4.2.0 [47]. To detect microbial taxa that varied significantly between non-rhizosphere and rhizosphere soils, we conducted Linear Discriminant Analysis (LDA) Effect Size (LEfSe) analysis using the “microbiomeMarker” package in R. The LDA discrimination threshold was set at 3.0, and a significance level of *p* < 0.05 was employed [48]. Microbial alpha diversity (e.g., goods_ coverage, Peilou, ACE, Chao1) and beta diversity (non-metric multidimensional scaling, NMDS) were estimated using the R “vegan” package [49]. The species composition of microbial communities was estimated using the R “amplicon” package [50]. A linear regression analysis was conducted to examine the impact of salinity on the alterations in microbial alpha diversity and ASV relative abundance using the “ggpmisc” package in R, and *p*-values were calculated to determine whether the results were reliable [51]. The PERMANOVA test was used to detect whether there were significant differences in community composition between groups [52].

To pinpoint the biotic and abiotic drivers of microbial diversity, a random forest model between different factors and alpha diversity was established using the R “rfPermute” package [53]. Spearman correlation analysis was used to determine whether factors had a significant correlation with bacterial alpha diversity [54]. To assess the impact of soil environmental factors on microbial species composition, the R package “linkET” was utilized to conduct the Mantel test [55]. The redundancy analysis (RDA) and canonical correspondence analysis (CCA) were conducted using the “vegan” package in R to investigate the impact of soil physicochemical factors on the structure of microbial communities [56].

The R “ICAMP” package was adopted to determine the primary mechanisms behind microbial community assembly in each sample group, and the R “NST” package was used to calculate the beta nearest taxon index (βNTI) of bacterial and fungal communities in non-rhizosphere and rhizosphere soils [57,58,59]. To characterize the responses of microbial co-occurrence networks to salt stress, the SparCC correlation of ASVs and *p*-values in different sample groups were calculated using the R “Fastspar” package [60]. Based on the INAP platform (https://inap.denglab.org.cn, accessed on 6 March 2023), the bipartite network of fungi and bacteria and the co-occurrences network of bacteria were constructed, and the network topological parameters were calculated [61]. Different colors were used to distinguish fungi and bacteria in the bipartite network (or major modules in the bacterial co-occurrence network) when visualizing all networks with the R “igraph” package [62]. Utilizing the “sub_graph” function from the R “igraph” package, we computed the topological parameters of bipartite subnetworks [62]. The topological changes in the non-rhizosphere bacterial network with salinity were analyzed based on linear regression models, and a *t*-test was used to determine the *p*-value for calculating the linear regression.

To provide a deeper understanding of how biotic and abiotic factors impact bacterial communities in distinct soil environments, we employed Partial Least Squares Path Modeling (PLS-PM) using the “plspm” package in R [63]. These models were constructed for both non-rhizosphere and rhizosphere communities. In each model, the soil properties comprised EC, pH, SWC, and SOM, and the soil nutrients consisted of TN, TP, TK, AK, AP, and AN. The fungal and bacterial richness is referred to as the Sobs index. The fungal and bacterial composition differences among samples were represented by the first axis of NMDS. The bacterial network consisted of the topological properties of each subnetwork, i.e., node number, network average degree, average path length, centrality betweenness, modularity, and vulnerability. The reliability of the models was determined by the goodness-of-fit (GOF) values, where the non-rhizosphere model achieved a GOF value of 0.73, and the rhizosphere model achieved a GOF value of 0.86. The bootstrapping method was used to calculate the significance of PLS-PM results [64].

## 3. Results

### 3.1. Contrasting Physicochemical Profiles of Non-Rhizosphere and Rhizosphere Soils

Significant differences (*p* < 0.05) between soils in the non-rhizosphere and rhizosphere were observed in the physicochemical indicators tested, including EC, SOM, pH, and AN (Figure 2a–d). In non-rhizosphere soil, the median, minimum, and maximum values of EC were 11.14 ds/m, 18.19 ds/m, and 3.16 ds/m, respectively, while in rhizosphere soil, they were 3.71 ds/m, 11.61 ds/m, and 0.99 ds/m, respectively. Similarly, for SOM in non-rhizosphere soil, the median, minimum, and maximum values were 16.06 g/kg, 5.63 g/kg, and 22.35 g/kg, whereas in rhizosphere soil, they were 11.31 g/kg, 5.63 g/kg, and 21.77 g/kg. The pH in non-rhizosphere soil had median, minimum, and maximum values of 7.6, 7.2, and 8.0, respectively, while in rhizosphere soil, the values were 7.4, 7.1, and 7.7, respectively. Finally, for AN in non-rhizosphere soil, the median, minimum, and maximum values were 254.8 mg/kg, 67.2 mg/kg, and 627.2 mg/kg, respectively, while in rhizosphere soil, they were 64.4 mg/kg, 22.4 mg/kg, and 232.4 mg/kg, respectively. In non-rhizosphere soil, there was a significant positive correlation between soil AN and EC, with a correlation coefficient of R^2^ = 0.34, *p* = 0.002 (Figure 2e). In rhizosphere soil, AN and SWC showed a significant positive correlation with EC. The correlation coefficients were R^2^ = 0.31 and R^2^ = 0.26, with *p* values of 0.004 and 0.009, respectively (Figure 2f).

### 3.2. Distinct Microbiomes in Non-Rhizosphere and Rhizosphere Soils Respond to Salt Stress

Soil salinity decreased the alpha diversity of soil bacteria, although this decrease was not pronounced in rhizosphere soil. In addition, the alpha diversity of fungi was less influenced by soil salinity. Linear regression analysis revealed significant negative correlations of soil bacterial Chao1 (R^2^ = 0.62, *p* < 0.01) and Shannon indices (R^2^ = 0.60, *p* < 0.01) against salinity (EC) in non-rhizosphere soil, whereas no significant correlations (*p* > 0.05) were observed in rhizosphere soil (Figure 3a,b). Similarly, no significant correlations were found for fungal Chao1/Shannon indices against salinity, both in non-rhizosphere and rhizosphere soil (*p* > 0.05) (Appendix A). LEfSe analysis identified twenty-three bacterial species (at the genus level) with significantly different relative abundances between non-rhizosphere and rhizosphere soil (*p* < 0.05, LDA > 3.0), including fourteen in rhizosphere (e.g., *Salinibacter*, *Salinimicrobium*, *Nafulsellwqa*) and nine in non-rhizosphere soil (e.g., *Microcoleus*, *Sphingomonas*, *Pseudomonas*) (Figure 4a). Additionally, 28 bacterial species showed significantly different relative abundances among different salinity levels in non-rhizosphere soil (*p* < 0.05, LDA > 3.2), with distinct taxa associated with increasing salinity levels. Similar trends were observed in rhizosphere soil, with 24 bacterial species showing significant differences (*p* < 0.05, LDA > 2.6) (Figure 4b,c). NMDS analysis further confirmed significant differences in bacterial community structures between non-rhizosphere and rhizosphere soil, as well as among different salinity levels within each soil type (Figure 3c,d and Figure 5a). In contrast, NMDS analysis suggested minor differences in fungal community structures between non-rhizosphere and rhizosphere soil, as well as among different salinity levels within each soil type (Figure 5b–d). Microbial composition in non-rhizosphere and rhizosphere soil responded differently to salinity. In terms of bacterial species composition (at the phylum level), Chloroflexi, Acidobateria, Saccharibacteria, Verrucomicrobia, and Nitrospira significantly decreased in relative abundance with increasing salinity in non-rhizosphere soil, while Bacteroidetes and Gemmatimonadetes increased (Figure 3e and Appendix A). In rhizosphere soil, higher salinity led to a decrease in Chloroflexi and Acidobacteria, accompanied by an increase in Bacteroidetes. Notably, Bacteroidetes became dominant in higher salinity conditions, while Proteobacteria lost their dominance (Figure 3e and Appendix A).

### 3.3. Various Factors Potentially Drive Microbiome Diversity in Saline Soils

The results obtained from the random forest model indicated that bacterial diversity in non-rhizosphere soil was jointly influenced by fungi and salinity. The vector of fungal composition differences among samples (NMDS1 axis) in non-rhizosphere soil exhibited significant correlations (*p* < 0.05) with alpha diversity indices of bacterial communities, including Coverage, Pielou, obs, Ace, Chao, Richness, Simpson, and Shannon indices, with the percentage of increase in mean square error (increase in MSE (%)) values reaching 20%. Moreover, EC showed significant correlations (*p* < 0.05) with alpha diversity indices of bacterial communities in non-rhizosphere soil, with MSE values reaching 20% for Coverage, Ace, Chao, Richness, and Shannon indices, and 10%, 15%, and 5% for Pielou, obs, and Simpson indices, respectively. Additionally, AN significantly influenced the Coverage and Ace indices of bacteria, with an MSE value of 10% (Figure 6a). In rhizosphere soil, bacterial diversity was significantly affected by fungi and total phosphorus (TP), with the influence of salinity comparatively less pronounced. The vector of fungal composition differences among samples (NMDS1 axis) showed significant correlations (*p* < 0.05) with alpha diversity indices of bacterial communities, with MSE values of 20% for Coverage, obs, Ace, Chao, Richness, and Shannon indices, and 10% for the Simpson index. TP also exhibited significant correlations (*p* < 0.05) with alpha diversity indices of bacterial communities, with MSE values of 20% for Coverage, obs, Ace, Chao, and Richness indices, and 10% and 15% for Simpson and Shannon indices, respectively (Figure 6b).

The Mantel analysis indicated that in non-rhizosphere soil, disregarding fungal influences, EC was the primary factor affecting bacterial species composition (ASV level) (R > 0.3, *p* < 0.01). Additionally, TP and AK also significantly influenced bacterial species composition (R > 0.1, *p* < 0.05) (Figure 6c). In rhizosphere soil, bacterial species composition was significantly influenced by pH (R > 0.1, *p* < 0.01), EC (R > 0.1, *p* < 0.05), and SWC (R > 0.1, *p* < 0.05) (Figure 6d). In non-rhizosphere soil, fungal species composition was solely affected by EC (R > 0.1, *p* < 0.05) (Figure 6c). Conversely, in rhizosphere soil, fungal species composition was influenced by multiple factors, including pH (R > 0.1, *p* < 0.05), EC (R > 0.1, *p* < 0.05), TP (R > 0.1, *p* < 0.05), and SWC (R > 0.1, *p* < 0.05) (Figure 6d).

The redundancy analysis revealed that in non-rhizosphere soil, EC emerged as the primary factor influencing bacterial community structure, with TK and AN also significantly affecting community structure (*p* < 0.05). These three factors exhibited positive correlations with the bacterial community in high-salinity soil (N4). Additionally, SWC also influenced bacterial community structure, positively correlating with the bacterial community structure in lower-salinity soil (N1) (Figure 6e). In rhizosphere soil, ranked by their influence magnitude, the significant soil factors affecting bacterial community structure were EC, AN, TP, TN, and AK (*p* < 0.05). Among these, EC, AN, TP, and AK were positively correlated with the community structure in high-salinity soil (R4), while TN was positively correlated with the community structure in low-salinity soil (R1) (Figure 6f). In non-rhizosphere soil, the significant soil factors influencing fungal community structure were EC, AN, SOM, and SWC, with EC, AN, and SOM positively correlating with the fungal community in high-salinity soil (N4) and SWC positively correlating with the fungal community structure in lower-salinity soil (N1) (Figure 6g). In rhizosphere soil, the significant soil factors influencing fungal community structure were EC, SWC, and AN, and all positively correlated with the fungal community in high-salinity soil (N4) (Figure 6h).

### 3.4. Salt Stress Coupled to Biotic Factors Shapes Microbiome Assembly

In non-rhizosphere soils, although the proportion of heterogeneous selection in bacterial community assembly increases with salt concentration, bacterial community assembly is primarily governed by deterministic selection in high-salinity environments. Infer community assembly mechanisms by phylogenetic bin-based null model analysis (ICAMP analysis) revealed that in non-rhizosphere soils at varying salinity levels (N1–N4), deterministic selection accounted for 44%, 51%, 67%, and 64% of bacterial community assembly, while heterogeneous selection accounted for 11%, 18%, 21%, and 32%, respectively (Figure 7a). In rhizosphere soils at different salinity levels (R1–R4), bacterial community assembly is predominantly driven by deterministic selection, with deterministic selection proportions increasing with salinity to 57%, 59%, 72%, and 75%, while heterogeneous selection proportions were 22%, 27%, 38%, and 42%, respectively (Figure 7b). For fungi, stochastic processes overwhelmingly govern community assembly in both non-rhizosphere and rhizosphere soils. In non-rhizosphere soils at different salinity levels (N1–N4), stochastic selection accounted for 100%, 100%, 99%, and 87% of community assembly, respectively (Figure 7c), while in rhizosphere soils at different salinity levels (R1–R4), stochastic selection accounted for 100%, 96%, 83%, and 75%, respectively (Figure 7d). It is noteworthy that although fungal community assembly is predominantly driven by stochastic processes, the proportion of homogeneous selection also increases with salinity, indicating a trend toward community structure stabilization with increasing salinity (non-rhizosphere N1–N4: 0%, 0%, 1%, 13%; rhizosphere R1–R4: 0%, 4%, 17%, 25%) (Figure 7c,d).

### 3.5. Salt Stress Destabilizes Microbiome Networks

Under salt stress, the abundance of fungi species was fewer compared to bacteria in both rhizosphere and non-rhizosphere soils, yet they held a more prominent position in the bipartite network (Figure 8a,b, Appendix A). In the bipartite networks of fungi and bacteria, the number of fungal species in non-rhizosphere and rhizosphere soils was 37 and 31, respectively. The average number of connections for fungi was 27.03 and 48.29 in non-rhizosphere and rhizosphere soils, respectively. Meanwhile, the average number of shared partners for fungi was 3.266 and 17.622 in rhizosphere and non-rhizosphere soils, respectively. Regarding bacteria, the number of species in non-rhizosphere and rhizosphere soils was 286 and 204, respectively. The average number of connections for bacteria was 3.497 and 7.338 in non-rhizosphere and rhizosphere soils, respectively, while the average number of shared partners for bacteria was 0.573 and 2.088 in non-rhizosphere and rhizosphere soils, respectively. Moreover, compared to the non-rhizosphere environment, fungi exhibited a higher level of interaction with bacteria in the rhizosphere.

The node degree of the bacterial co-occurrence network under different salinity levels in the rhizosphere exhibited a power law distribution (R^2^ > 0.97; Appendix A), indicating the scale-free and non-random characteristics of the network. As salinity increased, the size of the bacterial co-occurrence network decreased in non-rhizosphere soil, rendering the network more fragile. The number of nodes in the bacterial co-occurrence network decreased with increasing salinity levels (N1–N4: 779, 560, 460, 269) (Figure 8c–f, Appendix A). Linear regression results showed a significant negative correlation between salinity and both the total number of nodes and maximum stress centrality of the bacterial co-occurrence network (R^2^ = 0.98, 0.96, *p* < 0.01). Moreover, salinity exhibited a significant positive correlation with both modularity and vulnerability of the bacterial co-occurrence network (R^2^ = 0.90, 0.92, *p* < 0.05) (Figure 8g–j). In the rhizosphere network, the total number of nodes in the bacterial co-occurrence network did not exhibit a clear trend with different salinity levels (R1–R4: 103, 243, 65, 206) (Appendix A). Other topological attributes of the bacterial co-occurrence network under different salinity levels also did not show distinct patterns.

### 3.6. Biotic and Abiotic Factors Influence Microbiomes under Salt Stress

In this study, two Partial Least Squares Path Modeling (PLS-PM) models were used to understand how biotic and abiotic factors contribute to the formation of bacterial communities in saline soils. In non-rhizosphere soil, soil properties directly and negatively influenced bacterial composition differences among samples, whereas the bacterial co-occurrence network played a direct and positive role in both bacterial composition differences among samples and richness. The bacterial co-occurrence network exerted indirect impacts on the composition and richness of bacterial communities through the influence of soil properties on soil nutrients (Figure 9a). In rhizosphere soil, soil properties indirectly influence fungal composition differences among samples via soil nutrients. Further, fungal composition differences among samples had a significant influence on bacterial community composition and co-occurrence network, whereas bacterial richness was strongly shaped by the bacterial co-occurrence network and fungal richness (Figure 9b). The influence of the non-rhizosphere bacterial co-occurrence network on bacterial richness was stronger than that of the rhizosphere bacterial network.

## 4. Discussion

In saline environments, an increase in osmotic pressure leads to a decrease in species diversity, as species with lower salt tolerance are unable to survive, thus facilitating the dominance of salt-tolerant species [65]. This study has ascertained that salinity (represented by EC) is a major factor influencing bacterial species composition and community structure under salt stress, potentially through multiple mechanisms. In this study, the influence of salinity on the intricate dynamics governing soil nutrient fluxes, notably nitrogen, was critical in bacterial diversity and microbial cross-domain interactions [66]. Additionally, salinity also mediated heterogeneous selection in bacterial community assembly [67]. Furthermore, salinity reduced bacterial interactions and network size, promoting the transition of bacterial communities toward smaller modularization under salt stress. This observation is consistent with the results of previous studies that show that under increasing environmental stress, bacterial co-occurrence network modularity increases, resulting in more small modules that aid in resisting environmental stress [68].

Fungal–bacterial interactions in the bipartite network indicated that although fungi were less abundant in saline soils; they occupied a crucial ecological niche. Previous studies highlighted the higher connectivity of fungi compared to bacteria in bipartite networks, which are crucial for maintaining the stability of microbial ecosystems in high-altitude regions [69]. Therefore, fungi may play a key role in maintaining microbial diversity in saline soils. Soil fungal richness has been shown to regulate the balance of soil bacterial community construction processes, reducing stochastic assembly processes as fungal richness increases [54]. Fungi can influence the development of soil bacterial niches [70]. Combined with these findings, it was proposed that fungi can affect bacterial community assembly in saline environments, thereby influencing bacterial diversity. Fungi may act as hubs of microbial interactions to alleviate stress on ecosystems, representing a potential mechanism for maintaining ecosystem stability [71]. The stabilization of fungal community structure under salt stress serves as a stable “anchor” through which bacteria mitigate the effects of soil salinity.

Non-rhizosphere and rhizosphere soils exhibit different physicochemical conditions, with plant roots having lower salinity and pH, suggesting that the rhizosphere can alleviate environmental stress on microorganisms [72]. Relative to bulk soils, bacterial alpha diversity and community structure in rhizosphere soils did not significantly change with increasing salinity. One of the underlying mechanisms might be that plant roots produce and release exudates and signaling molecules into the rhizosphere, which can induce community succession and increase network stability [73]. Furthermore, the topological properties of bacterial co-occurrence networks remained stable in rhizosphere soil under different salinity levels. PLS-SEM results further showed a closer relationship between fungal and bacterial communities in the rhizosphere. It has been acknowledged that rhizosphere fungi have higher diversity and more stable co-occurrence networks [74,75]. Many arbuscular mycorrhizal fungi act as extensions of plant roots and recruit bacteria, which can contribute to community drought resistance [76]. Therefore, the rhizosphere could possibly buffer harsh conditions for microorganisms by enhancing fungal–bacterial interactions and, consequently, alleviating salt stress on bacterial communities.

## 5. Conclusions

This study showed that higher soil salinity could result in decreased bacterial diversity, altered community structure, and lowered co-occurrence network complexity and stability. However, fungal diversity was less affected by soil salinity. Fungal richness and community structure might regulate bacterial diversity and community assembly under salt stress. Enhanced interactions between bacteria and fungi occurred in the rhizosphere. The results of this study underscore the role of fungal–bacterial interactions under salt stress in the rhizosphere to diminish the adverse effects of environmental stress on microbiome function.

## Figures and Tables

**Figure 1 microorganisms-12-00683-f001:**
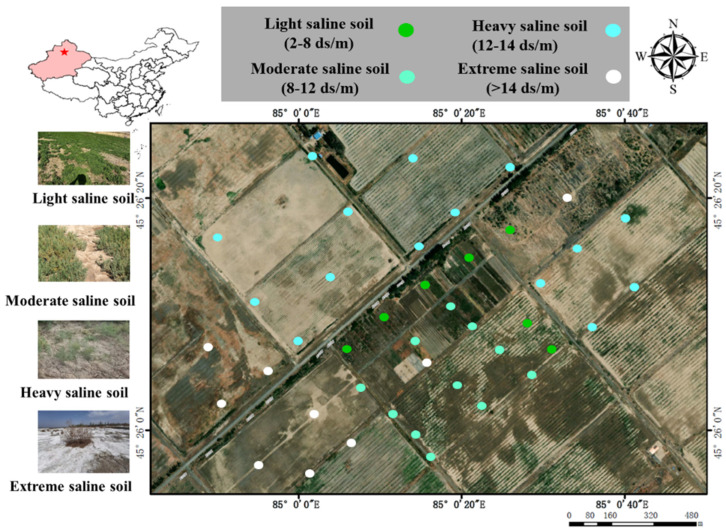
Geological setting and geographical distribution of sampling points within a salt gradient. The red star indicates the location of the sampling point on the map.

**Figure 2 microorganisms-12-00683-f002:**
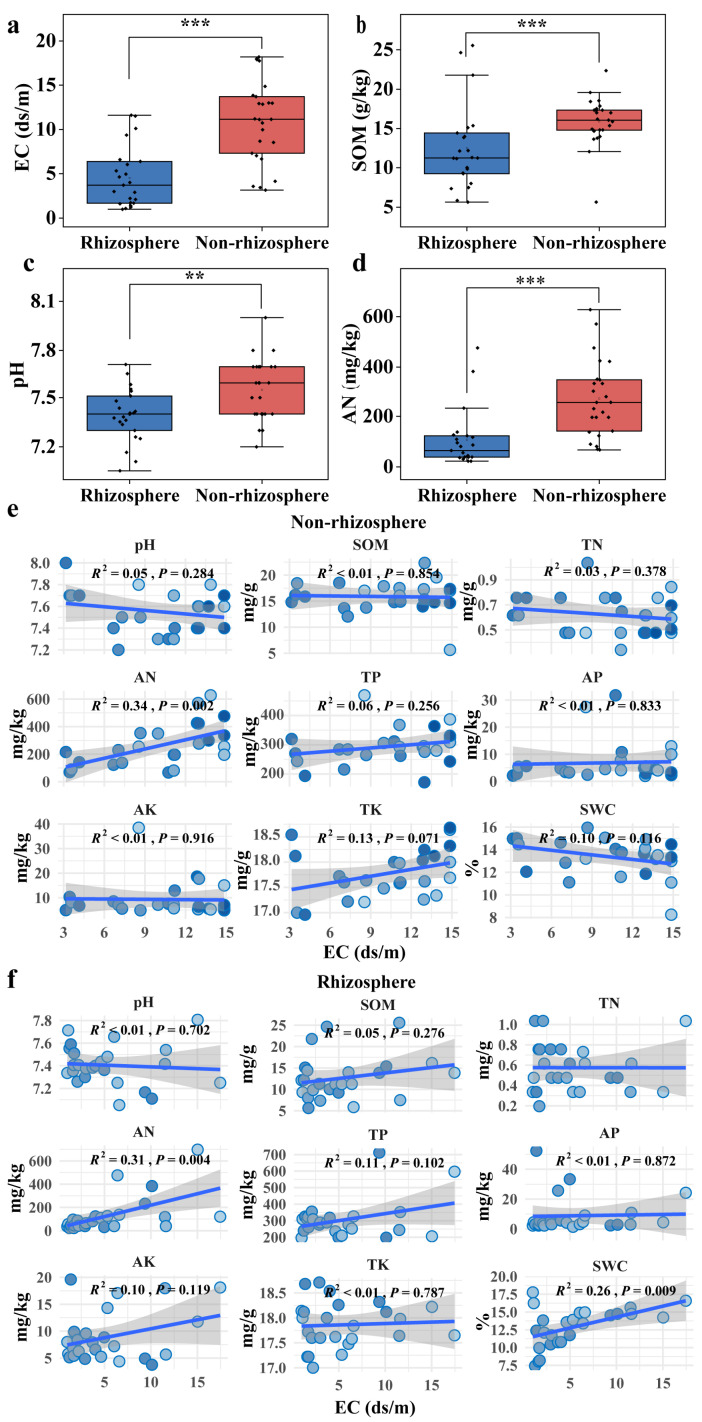
Differences in the physicochemical properties between non-rhizosphere and rhizosphere soils: (**a**) EC; (**b**) SOM; (**c**) pH; and (**d**) AN. ** *p* < 0.01 and *** *p* < 0.001. The relationship between salinity and other physicochemical properties in (**e**) non-rhizosphere soil and (**f**) rhizosphere soil based on linear regression analysis. EC: electrical conductivity, SOM: soil organic matter, TN: total nitrogen, AN: available nitrogen, TP: total phosphorus, AP: available phosphorus, TK: total potassium, AK: available potassium, and SWC: soil water content. The gray represents the confidence interval, the blue line represents the linear regression line, and the balls of different colors represent different samples.

**Figure 3 microorganisms-12-00683-f003:**
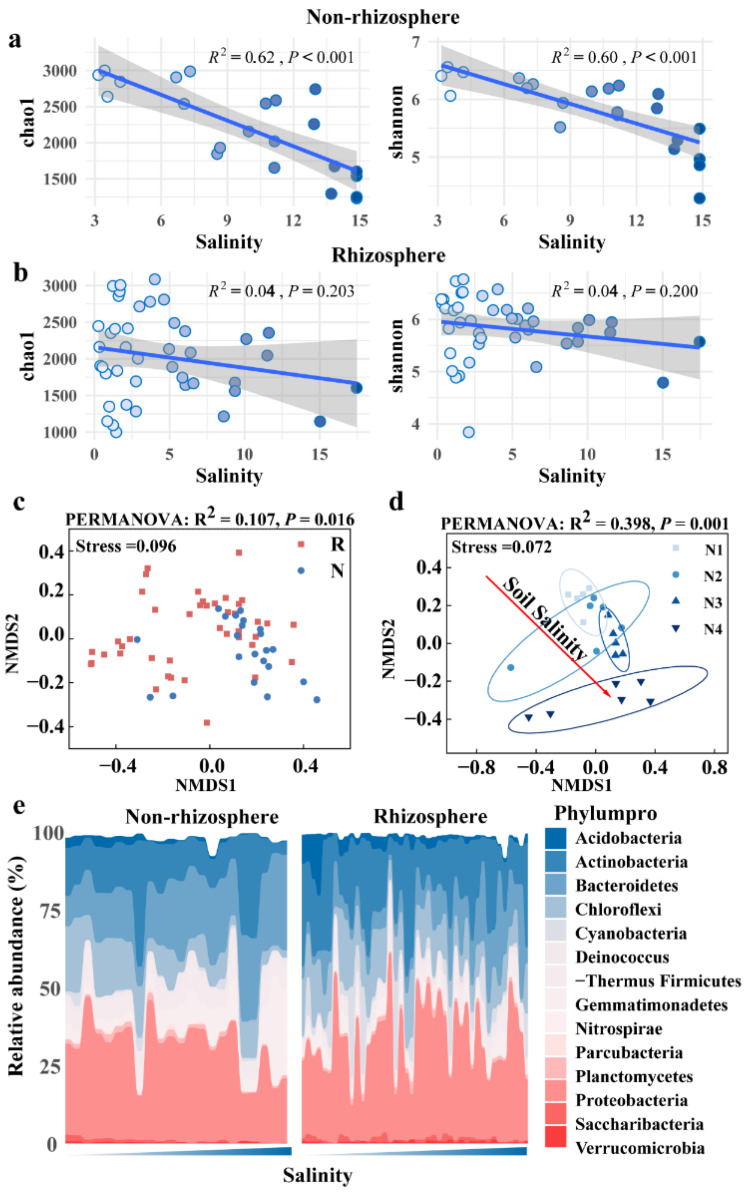
Differences in bacterial communities between non-rhizosphere and rhizosphere soils: the relationship between alpha diversity and salinity revealed by linear regression in (**a**) non-rhizosphere soil and (**b**) rhizosphere soil, the gray represents the confidence interval, the blue line represents the linear regression line, and the balls of different colors represent different samples; and (**c**) bacterial community structure visualized by NMDS based on Bray–Curtis dissimilarities between non-rhizosphere and rhizosphere soils. N means non-rhizosphere soil, R means rhizosphere soil; and (**d**) is the response of bacterial community structure to salinity level in non-rhizosphere soil. N1–N4 represents non-rhizosphere soils with different salinities the ellipse represents the confidence interval ellipse for each group. The specific divisions are shown in Appendix A; (**e**) changes in taxa (at the phylum level) relative abundance with salinity.

**Figure 4 microorganisms-12-00683-f004:**
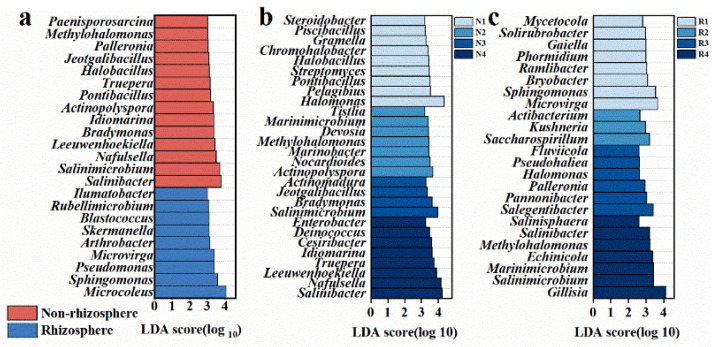
Major differentially abundant taxa (LDA threshold 3.0, *p* < 0.05) between non-rhizosphere and rhizosphere soils (**a**) and major differentially abundant taxa (LDA threshold 3.0, *p* < 0.05) between different salinity levels in (**b**) non-rhizosphere and (**c**) rhizosphere soils. The different salinity levels in non-rhizosphere soil are classified as N1–N4, while the different salinity levels in rhizosphere soil are classified as R1–R4. The specific criteria for this classification are shown in Appendix A.

**Figure 5 microorganisms-12-00683-f005:**
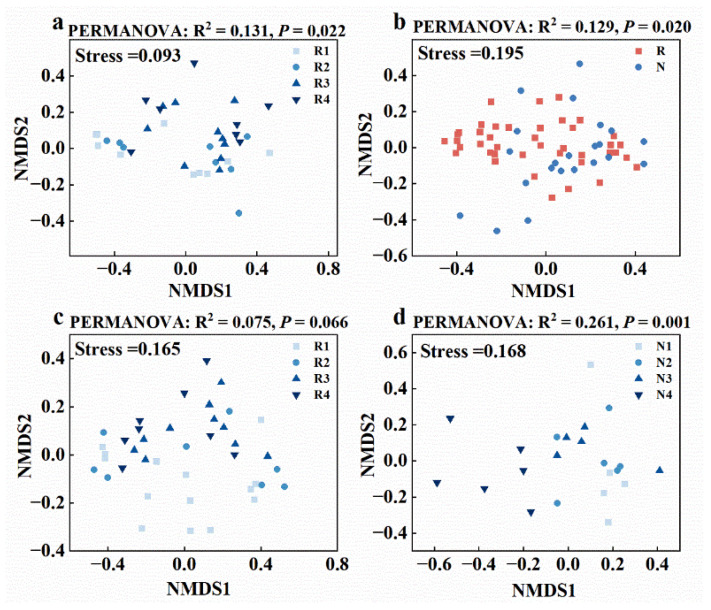
Rhizosphere bacterial community structure under different salinity levels: (**a**) differences in (**b**) fungal community structure between non-rhizosphere and rhizosphere soils; the response of fungal community structure under different salinity levels in (**c**) rhizosphere and (**d**) non-rhizosphere soils. N means non-rhizosphere soil, R means rhizosphere soil, and the different salinity levels in non-rhizosphere soil are classified as N1–N4, while the different salinity levels in rhizosphere soil are classified as R1–R4. The specific criteria for this classification are shown in Appendix A.

**Figure 6 microorganisms-12-00683-f006:**
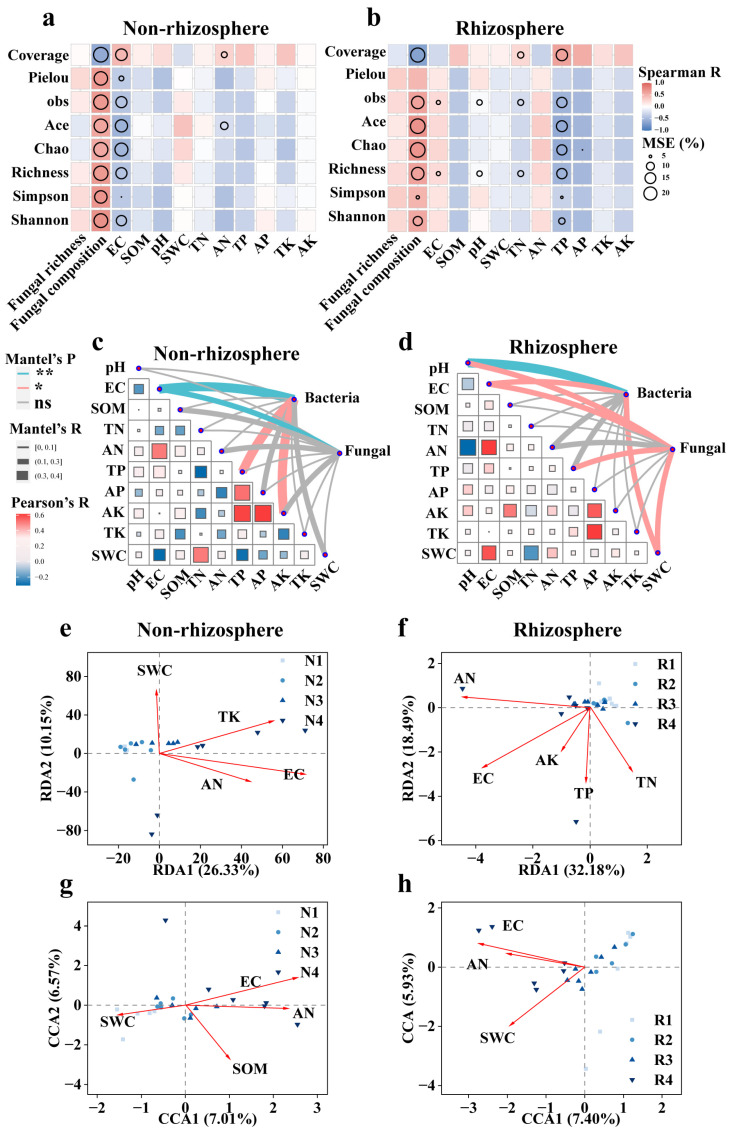
Effects of biotic and abiotic factors on bacterial and fungal communities in saline soils. The contributions of various factors to bacterial alpha diversity in (**a**) non-rhizosphere and (**b**) rhizosphere soils predicted by correlation and random forest models. The size of the circles indicates the importance of the variables based on the percentage of increase in mean square error (increase in MSE (%); only those with *p* < 0.05 are retained). The color gradient indicates the Spearman correlation coefficient. The effects of soil environmental factors on bacterial community composition in (**c**) non-rhizosphere and (**d**) rhizosphere soils, assessed by the Mantel test. The color and thickness of the lines indicate the significance and correlation coefficient of the Mantel test, respectively; the color gradient and block size indicate the Pearson correlation coefficient. Major soil factors affecting bacterial and fungal beta diversity in (**e**,**g**) non-rhizosphere and (**f**,**h**) rhizosphere soils identified by RDA and CCA, respectively (only significant factors with *p* < 0.05 are retained). The different salinity levels in non-rhizosphere soil are classified as N1–N4, while the different salinity levels in rhizosphere soil are classified as R1–R4. The specific criteria for this classification are shown in Appendix A.

**Figure 7 microorganisms-12-00683-f007:**
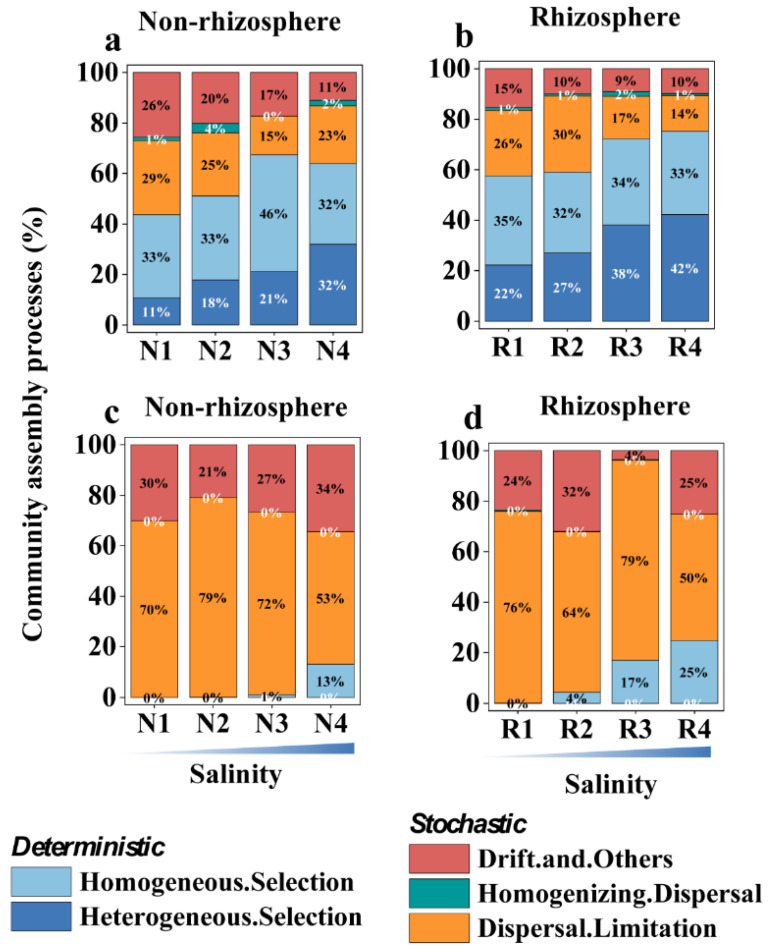
Effects of biotic and abiotic factors on microbial community assembly in saline soils. The proportion of community assembly processes for soil bacteria in (**a**) non-rhizosphere and (**b**) rhizosphere soils and fungi in (**c**) non-rhizosphere and (**d**) rhizosphere soils under different salinity levels. The different salinity levels in non-rhizosphere soil are classified as N1–N4, while the different salinity levels in rhizosphere soil are classified as R1–R4. The specific criteria for this classification are shown in Appendix A.

**Figure 8 microorganisms-12-00683-f008:**
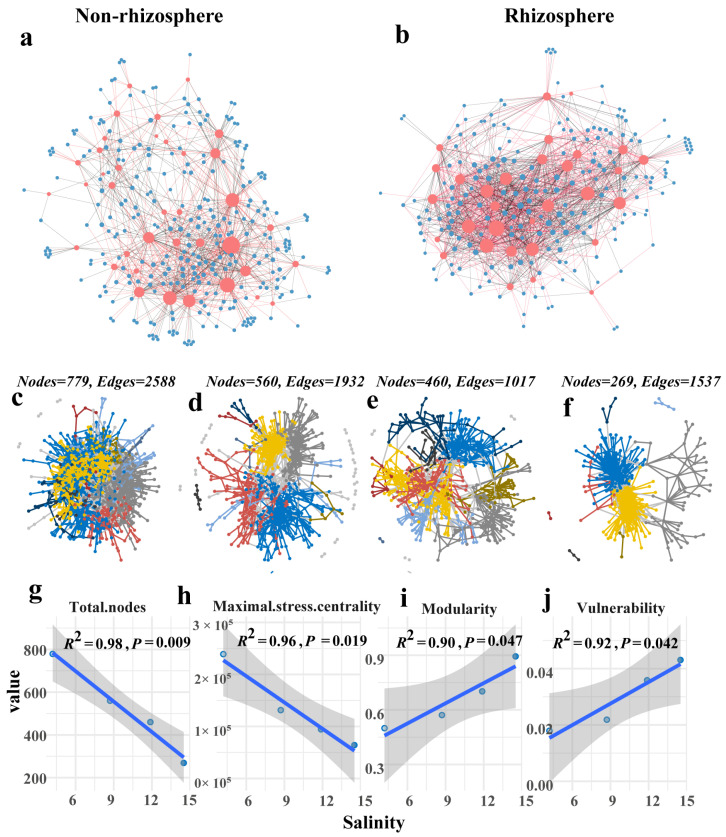
Bipartite and co-occurrence networks of soil microbial communities under salt stress. The bipartite networks in (**a**) non-rhizosphere and (**b**) rhizosphere soils, where the nodes are colored blue (red) for bacteria (fungi), the node size indicates the degree, and the red (black) links indicate positive (negative) correlations. The evolution of the bacterial co-occurrence network structure under different salinity levels in non-rhizosphere soil ((**c**): N1, (**d**): N2, (**e**): N3, (**f**): N4), where the nodes and links are colored according to module attributes. (**g**–**j**) The changes in bacterial co-occurrence network topology with salinity in non-rhizosphere soil (**g**): total nodes, (**h**): maximal stress centrality, (**i**): modularity, (**j**): vulnerability), the gray represents the confidence interval, the blue line represents the linear regression line, and the balls of different colors represent different samples. The different salinity levels in non-rhizosphere soil are classified as N1–N4. The specific criteria for this classification are shown in Appendix A.

**Figure 9 microorganisms-12-00683-f009:**
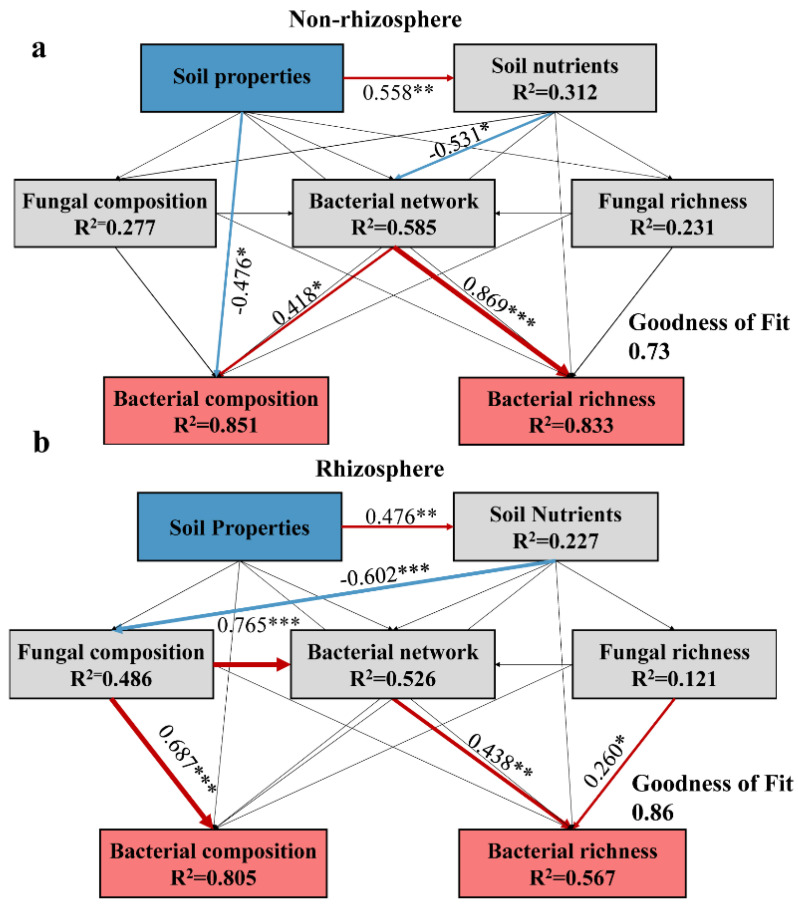
PLS-PM revealed the impacts of different factors on bacterial communities in non-rhizosphere (**a**) and rhizosphere (**b**) soils. PLS-PM described the relationships among soil properties (EC, pH, SWC, and SOM), soil nutrients (TN, TP, TK, AK, AP, and AN), fungal composition (fungal composition differences among samples (NMDS1 axis)), bacterial network (topological properties of each subnetwork, i.e., node number, network average degree, average path length, centrality betweenness, modularity, and vulnerability), fungal richness (fungal Sobs index) in relation to bacterial composition (bacterial composition differences among samples (NMDS1 axis)), and bacterial richness (bacterial Sobs index) in saline soils. The red and blue arrows reflect positive and negative relationships, respectively, with widths proportional to the strengths. The solid black arrows indicate non-significant relationships. The numbers beside the arrows are the standardized path coefficients. * *p* < 0.05, ** *p* < 0.01, and *** *p* < 0.001.

## Data Availability

Data will be made available upon request.

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
