# Peer review of "Understanding Salinity-Driven Modulation of Microbial Interactions: Rhizosphere versus Edaphic Microbiome Dynamics"

_microorganisms, 2024, doi:10.3390/microorganisms12040683_

Round 1

Reviewer 1 Report

Comments and Suggestions for Authors

The reviewed research may constitute a valuable addition to knowledge regarding the biology and ecology of the rhizosphere. Information about microorganisms may be of not only theoretical but also practical importance in understanding the importance of biodiversity in ecosystems.

The objectives of the research are clearly presented, although formulating them in the form of hypotheses could strengthen their significance, but this is only a suggestion and not an objection.

Please explain any abbreviations in the captions under Figures

In Figure 9, the gray arrows are faintly visible, giving the false impression of a lack of connections. I suggest a different color for the arrows

The conclusions were formulated correctly.

Author Response

  1. Please explain any abbreviations in the captions under Figures

Response: Abbreviations are now explained in the captions under Figures.

In the caption of Figure 2, EC, SOM, TN, AN, TP, AP, TK, AK, and SWC were explained as following. EC: electrical conductivity, SOM: soil organic matter, TN: total nitrogen, AN: available nitrogen, TP: total phosphorus, AP: available phosphorus, TK: total potassium, AK: available potassium, and SWC: soil water content.

In the caption of Figure 3, R, N, and N1-N4 standed for rhizosphere soil, non-rhizosphere soil, and non-rhizosphere soils with different salinities, respectively.

In the caption of Figure 4, 5, 6, 7, and 8, the meaning of abbreviations was explained as mentioned above.

  1. In Figure 9, the gray arrows are faintly visible, giving the false impression of a lack of connections. I suggest a different color for the arrows.

Response: We have now replaced gray arrows with solid black lines. We have also made corresponding changes in the caption of Figure 9.

Reviewer 2 Report

Comments and Suggestions for Authors

I really enjoyed reading this work. Just some comments and suggestions:

Introduction:

- Maybe too short and very focused. Requires some more context on microbial effects and ecology of saline environments, or soil impact that also affects the dynamics

Material and Methods:

- Check some minor errors (as Line 81)

- The 'standard practice' could be not well-known outside of China. In order to help the general understanding, I would suggest extending a little how to proceed here

- Despite bioinformatic approaches are quite clear, other tests and determinations require some more details to contrast the reliability of the work

Results:

- In general, the descriptive approach is too extensive. I would recommend to point only the main results, at least in the most relevant items. Most of the info is already in tables and graphs, so I suggest using the text to highlight those relevant differences

Conclusions:

- Remember in silico approaches needs to be validates, so despite the correlations, these results need to be validated by properly addressed testing to use the term as "demonstrated". You may want to consider to mild the assessments in this section

Author Response

1.Introduction:

Maybe too short and very focused. Requires some more context on microbial effects and ecology of saline environments, or soil impact that also affects the dynamics

Response: Thanks for your comments. The following content has been included in the Introduction:

High-salinity saline soil inhibits the relative abundance of bacteria and functional genes, affecting soil carbon metabolism, reducing soil carbon accumulation, and weakening soil carbon sink capacity. Soil salinization decreases the abundance of certain bacteria, significantly hinders the diversity of soil microorganisms, and impairs soil microbial functions. The soil microbiome, which plays an essential role in maintaining regional ecosystem health and stability, is sensitive to environmental stress. Changes in the soil microbiome in turn affect the functionality of the whole ecosystem.

  1. Material and Methods:

- Check some minor errors (as Line 81)

Response: Done.

  1. The 'standard practice' could be not well-known outside of China. In order to help the general understanding, I would suggest extending a little how to proceed here.

Response: In order to enhance the clarity of the soil property detection method, we have provided additional details and replaced

“According to the national standards of practice, the analyses of soil organic matter (SOM), total nitrogen (TN), total phosphorus (TP), total potassium (TK), available ni-trogen (AN), available phosphorus (AP), and available potassium (AK) were conducted [30].”

with

“Soil Organic Matter (SOM) in soil samples was quantified using the loss on ignition method [13]. Total soil nitrogen (TN) was measured using the Modified Kjeldahl method [36]. Available nitrogen (AN) was determined using an AA3 continuous-flow analysis system (SEAL Autoanalyzer 3hr, SEAL analytical Inc.) [37]. Total phosphorus (TP) was determined by alkali fusion–Mo-Sb Anti spectrophotometric method [38]. Available phosphorus (AP) was determined by Sodium hydrogen carbonate solu-tion-Mo-Sb anti spectrophotometric method [39]. Total potassium (TK) was deter-mined by acid-soluble atomic emission photometric method [40]. Available potassium (AK) was determined by flame photometer method [41].”

  1. Despite bioinformatic approaches are quite clear, other tests and determinations require some more details to contrast the reliability of the work

Response: We have refined the testing procedures within the statistical methods shown below:

1) The “ggpmisc” package in R was used to perform linear regression analysis and conduct a t test to determine the significance of the results;

2) The PERMANOVA test was used to detect whether there were significant differences in community composition between groups;

3) Spearman correlation analysis was used to determine whether factors had a significant correlation with bacterial alpha diversity;

4) Bootstrapping method is used to calculate the significance of PLS-PM results.

  1. In general, the descriptive approach is too extensive. I would recommend to point only the main results, at least in the most relevant items. Most of the info is already in tables and graphs, so I suggest using the text to highlight those relevant differences.

Response: Given the nature of the big data volume of this manuscript, we intend to detail the results in the context to help the readers understand the information in the tables and graphs.

  1. Remember in silico approaches needs to be validates, so despite the correlations, these results need to be validated by properly addressed testing to use the term as "demonstrated". You may want to consider to mild the assessments in this section.

Response: We have now rephrased the section with a neutral and moderate tone.

Reviewer 3 Report

Comments and Suggestions for Authors

The manuscript entitled 'Understanding Salinity-Driven Modulation of Microbial Interactions: Rhizosphere Versus Edaphic Microbiome Dynamics’ is well written, easy to follow, and in my opinion is almost ready for publication in the Journal of 'Microorganisms'.

When reading the work, I only had a few comments:

- Do the authors believe that the EC parameter alone is sufficient to characterize salinity? In this type of research, the content of individual elements, such as Na and Cl, could also be taken into account. Soil saturation with the basic cations Ca2+, K+, Mg2+ and Na+ could also be taken into account. Disturbances in the ratios of these ions also affect the rhizosphere zone of plants, and thus probably also modify the occurrence of specific species of bacteria and fungi.

- In the case of saline soils, their strong alkalization also occurs. Do the authors definitely claim that the results obtained are the result of excessive soil salinity?

-Under natural conditions, we are dealing with high variability of surrounding factors. Do the authors think that such an experiment can be carried out in controlled/laboratory conditions to confirm the obtained research results with certainty?

The above comments do not reduce the substantive value of the manuscript and are of a debatable nature. Due to the large number of presented research results, a very good presentation of the obtained results, and description of the experiment, I believe that the article is almost ready for publication.

Minor remarks:

-Lines 114-117 – please define exactly the methods utilized.

-All units used in the manuscript should be presented in the SI system

Author Response

  1. Do the authors believe that the EC parameter alone is sufficient to characterize salinity? In this type of research, the content of individual elements, such as Na and Cl, could also be taken into account. Soil saturation with the basic cations Ca2+, K+, Mg2+ and Na+ could also be taken into account. Disturbances in the ratios of these ions also affect the rhizosphere zone of plants, and thus probably also modify the occurrence of specific species of bacteria and fungi.

Response: It is a convenient method using EC to represent salinity in most studies. For example, Farzamian et al., 2023 Agricultural Water Management. Nevertheless, as mentioned by the reviewer that Ca2+, K+, Mg2+, and Na+ are the primary ions that make up soil salinity. We have carried out experiments to investigate the specific impacts of individual salt ions on soil microorganisms and the manuscript is under preparation.

  1. In the case of saline soils, their strong alkalization also occurs. Do the authors definitely claim that the results obtained are the result of excessive soil salinity?

Response: The relationship between soil salinization and soil alkalization involves numerous interactions between anions and cations. While our research has not yet focused on this interaction, we have conducted correlation and regression analyses which revealed no significant relationship between soil salinity and pH in the sampling area. We believed that soil salinization did not lead to soil alkalization in the sampling area.

  1. Under natural conditions, we are dealing with high variability of surrounding factors. Do the authors think that such an experiment can be carried out in controlled/laboratory conditions to confirm the obtained research results with certainty?

Response: To further validate our hypothesis, we have collected in-situ soil for laboratory experiments and developed strategies to improve saline soil. The results of these experiments are currently being compiled.

  1. Lines 114-117 – please define exactly the methods utilized.

Response: We have provided additional details and replaced

“According to the national standards of practice, the analyses of soil organic matter (SOM), total nitrogen (TN), total phosphorus (TP), total potassium (TK), available ni-trogen (AN), available phosphorus (AP), and available potassium (AK) were conducted [30].”

with

“Soil Organic Matter (SOM) in soil samples was quantified using the loss on ignition method [13]. Total soil nitrogen (TN) was measured using the Modified Kjeldahl method [36]. Available nitrogen (AN) was determined using an AA3 continuous-flow analysis system (SEAL Autoanalyzer 3hr, SEAL analytical Inc.) [37]. Total phosphorus (TP) was determined by alkali fusion–Mo-Sb Anti spectrophotometric method [38]. Available phosphorus (AP) was determined by Sodium hydrogen carbonate solu-tion-Mo-Sb anti spectrophotometric method [39]. Total potassium (TK) was deter-mined by acid-soluble atomic emission photometric method [40]. Available potassium (AK) was determined by flame photometer method [41].”

  1. All units used in the manuscript should be presented in the SI system

Response: We have now revised all units according to the International System of Units.